# Adolescents' reasons for accessing their health records online, perceived usefulness and experienced provider encouragement: a national survey in Sweden

Josefin Hagström  ,[1,2] Charlotte Blease,[1,3] Isabella Scandurra,[4] Jonas Moll,[4] Åsa Cajander,[5] Hanife Rexhepi,[6] Maria Hägglund[1,2]

For numbered affiliations see end of article.

**Correspondence to**
Josefin Hagström; josefin.hagstrom@uu.se

## ABSTRACT

**Background** Having online access to electronic health records (EHRs) may help patients become engaged in their care at an early age. However, little is known about adolescents using patient portals. A national survey conducted within the Nordic eHealth project NORDeHEALTH provided an important opportunity to advance our understanding of adolescent users of patient portals. The present study explored reasons for reading the EHRs, the perceived usefulness of information and functions in a patient portal and the association between frequency of use and encouragement to read the EHR.

**Methods** Data were collected in a survey using convenience sampling, available through the Swedish online health portal during 3 weeks in January and February 2022. This study included a subset of items and only respondents aged 15–19. Demographic factors and frequencies on Likert-style questions were reported with descriptive statistics, while Fisher's exact test was used to explore differences in use frequency based on having been encouraged to read by a healthcare professional (HCP).

**Results** Of 13 008 users who completed the survey, 218 (1.7%) were unique users aged 15–19 (females: 77.1%). One-fifth (47/218, 21.6%) had been encouraged by HCPs to read their records, and having been encouraged by HCPs was related to higher use frequency (p=0.018). All types of information were rated high on usefulness, while some functions were rated low, such as blocking specific clinical notes from HCPs and managing services for family members. The main reason for reading their health records online was out of curiosity.

**Conclusions** Adolescents who read their records online perceive it to be useful. Encouragement by HCPs can lead to increased use of patient portals among adolescents. Findings should be considered in the future design of patient portals for adolescents.

## INTRODUCTION

An increasing number of healthcare providers are giving patients access to their electronic health records (EHR), including patient information such as clinical notes, diagnoses and medication lists. The practice

### WHAT IS ALREADY KNOWN ON THIS TOPIC

⇒ Implementation of online record access for adolescents is controversial.
⇒ Citizens in Sweden have had online record access via a national patient portal for a decade.
⇒ Prior research has focused on adolescents without experience of using a patient portal, while existing studies of adolescent patient portal users have been limited to contexts of serious illness.
⇒ Adolescents report similar benefits of reading electronic health records (EHRs) to those of adults, such as reduced anxiety, enhanced knowledge about their illness, ability to ask informed questions and reflecting more on their health.

### WHAT THIS STUDY ADDS

⇒ Encouragement by healthcare professionals (HCPs) was related to higher use frequency. However, adolescents are rarely encouraged by HCPs to read their notes online.
⇒ Adolescents reading their records perceive all types of information in the EHR included in the survey as useful.

### HOW THIS STUDY MIGHT AFFECT RESEARCH, PRACTICE OR POLICY

⇒ Enhanced encouragement and support for adolescents will be required to support the transition into adulthood and involve adolescents in their own care.
⇒ EHR vendors designing patient portals and patient portals intended for adolescent use in particular may benefit from considering the types of information and functions that adolescents find useful.

of enabling access to free-text notes written in the EHRs by healthcare professionals (HCPs) is sometimes called 'open notes'.[1] Commonly, patients gain online record access (ORA) via secure online portals. Patient-accessible EHRs (PAEHR) are becoming increasingly expected among patients, even affecting patients' choice of healthcare provider. In

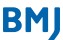

particular, adolescents are growing up with expectations of digital convenience that also applies to healthcare.[2]

During early childhood, PAEHR portal systems can often allow for parental proxy access, where parents or guardians of the minor have access to the EHR of the child. However, adolescents needing healthcare face a hidden battle in balancing privacy and personal autonomy with the need for support from parents. Worries about sensitive health information (eg, mental health issues and sexual health) being exposed to parents through the PAEHR can dissuade young people from accessing medical care.[3 4] The transition from adolescent to adult patienthood has therefore raised concerns among HCPs[5–7] and implementing online access to EHRs is especially controversial in this population.

The e-service for PAEHRs in Sweden is *Journalen*, available via the national web patient portal 1177.se, accessible to patients in all 21 Swedish regions.[8] According to publicly available statistics,[9] *Journalen* is being accessed by a steadily increasing number of users; approximately 1 360 000 unique users per month during 2022. Adult parent users have automatic access to their child's records in *Journalen*, but lose this access when the child turns 13,

and adolescents gain their record access at 16. When an adolescent is between 13 and 15 years old, parents and adolescents can apply for extended access outside of the default. Adolescent patients access the same information as adults. The content includes, for example, a list of prescribed medications, laboratory results, growth curves, vaccinations, diagnoses, maternity care records, referrals, warnings and treatment plans (figure 1). However, availability differs according to region and connected healthcare providers (who have agreed to give access) (figure 2). For example, psychiatric notes are shown in 17 of 21 regions.[10] Functions such as prescription renewal and appointment booking are conducted on the patient portal, yet externally from the PAEHR.

Two literature reviews on paediatric patient portals found limited studies in this age group. One reviewed definition and utilisation[11] while the other focused on use, views and experiences among adolescents.[12] While expressing a lack of knowledge of how to use PAEHRs,[2 13] adolescents have stated a desire to receive such information from HCPs.[2 14] In a US study examining EHR access of adolescents' records by accounts divided into self, surrogate and delegate, there was a significant increase

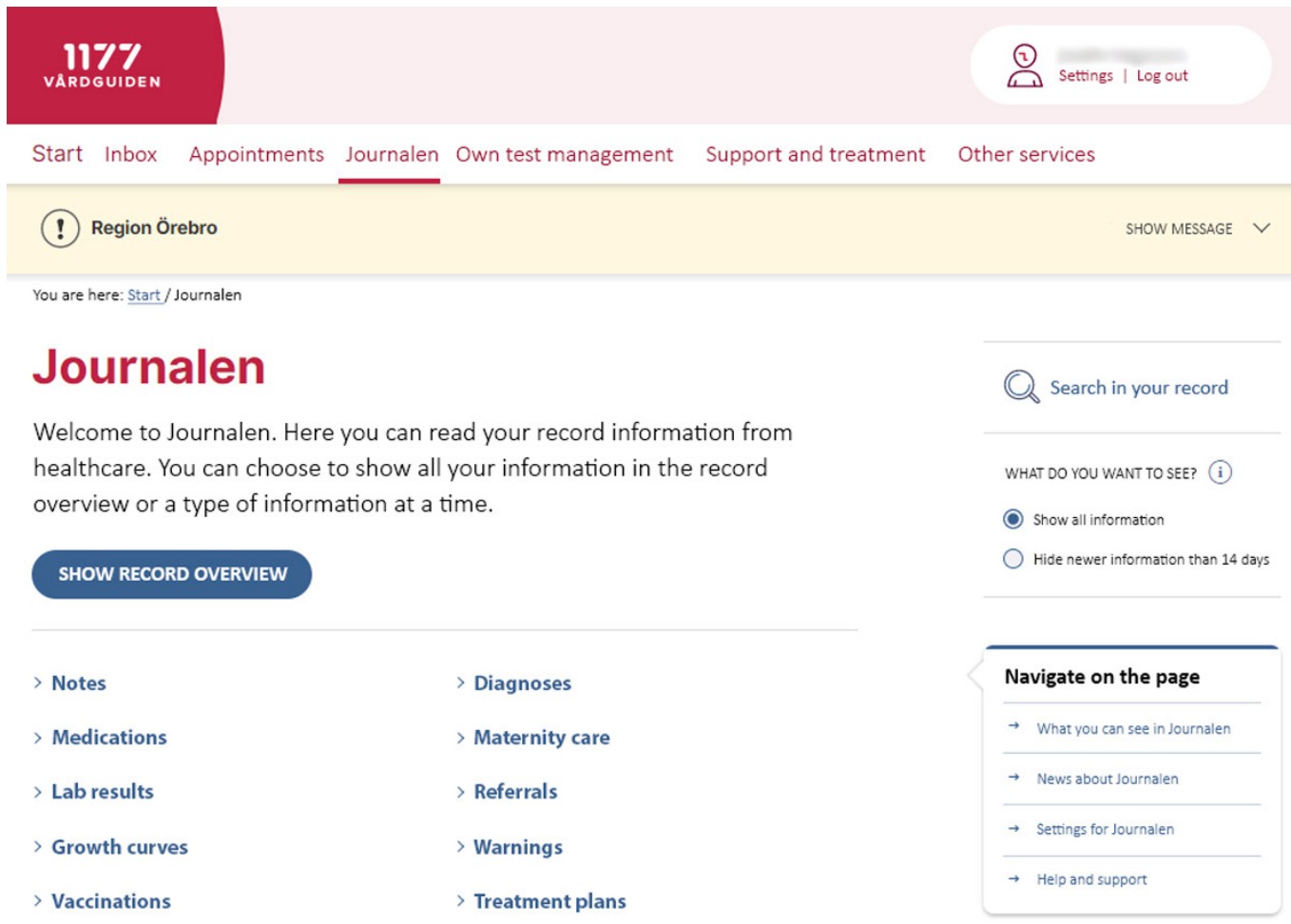

**Figure 1** The patient-accessible electronic health record *Journalen* after log-in (manually translated), showing the functions and information available.[39] Service produced by Inera under the auspices of Swedish municipalities and regions.

| | Medical Notes | Diagnoses | Lab results | Medications | Immunizations | Referrals | Medical alerts | Access Log Lists | Forms | Blocking of Record | Health contacts | Maternity care | Growth | Care plan | Total |
|---|---|---|---|---|---|---|---|---|---|---|---|---|---|---|---|
| Jönköping | | | | | | | | | | | | | | | 11 |
| Uppsala | | | | | | | | | | | | | | | 11 |
| Kronoberg | | | | | | | | | | | | | | | 10 |
| Värmland | | | | | | | | | | | | | | | 10 |
| Dalarna | | | | | | | | | | | | | | | 9 |
| Jämtland/Härjedalen | | | | | | | | | | | | | | | 9 |
| Kalmar | | | | | | | | | | | | | | | 9 |
| Södermanland | | | | | | | | | | | | | | | 9 |
| Östergötland | | | | | | | | | | | | | | | 9 |
| Blekinge | | | | | | | | | | | | | | | 8 |
| Norrbotten | | | | | | | | | | | | | | | 8 |
| Västernorrland | | | | | | | | | | | | | | | 8 |
| Västmanland | | | | | | | | | | | | | | | 8 |
| Västra Götaland | | | | | | | | | | | | | | | 8 |
| Örebro | | | | | | | | | | | | | | | 8 |
| Gävleborg | | | | | | | | | | | | | | | 7 |
| Gotland | | | | | | | | | | | | | | | 6 |
| Halland | | | | | | | | | | | | | | | 6 |
| Stockholm | | | | | | | | | | | | | | | 6 |
| Västerbotten | | | | | | | | | | | | | | | 6 |
| Skåne | | | | | | | | | | | | | | | 5 |
| Capio (private care provider) | | | | | | | | | | | | | | | 5 |
| Total | 22 | 20 | 21 | 15 | 18 | 12 | 13 | 5 | 1 | 20 | 21 | 4 | 3 | 1 | |

**Figure 2** The information shown in *Journalen* depending on region or healthcare provider (green squares) during the time of the survey.

of use during adolescence.[15] Furthermore, HCPs have stated the benefits of guiding adolescent patients to become more involved and activated in their care.[16 17] Research has mostly been observational or focused on adolescents' attitudes towards patient portals, rather than experiences. In observational studies, adolescents demonstrate the use of the same portal features as adults,[18] including laboratory results,[15 19 20] appointment review,[15 19 20] messaging[15 19] and medications.[20] Existing studies of adolescent users have been limited to contexts of serious illness, and these adolescents report similar benefits of reading EHRs to those of adults, such as reduced anxiety, enhanced knowledge about their illness, ability to ask informed questions and reflecting more on their health,[21] and empowerment.[5 21] A benefit unique for adolescents concerned a supported transition into adulthood.[21] To date, no studies have systematically examined how adolescents more broadly experience ORA.

## Study aim

A recent scoping review concluded that with few exceptions (Australia, Canada, New Zealand and the UK), the entire body of literature on adolescent access to EHRs had been carried out in the USA.[12] The present research explored reasons for reading the EHRs, the perceived usefulness of information and functions in a patient portal, and the association between frequency of use and HCP encouragement to read the EHR, among adolescents in Sweden.

## METHODS
### Study design

Within the NORDeHEALTH research project,[22] an open survey designed to elicit the opinions and experiences of PAEHR users was conducted. The data were collected for 3 weeks from January to February 2022. Convenience sampling was used. Participants were notified about the survey when they logged into their records through the Swedish national PAEHR *Journalen*, by receiving a request to voluntarily participate in the survey, along with detailed information about the study. Accessing the survey without logging in was impossible. The participation request did not appear once the patient had completed the survey. Participants provided electronic consent and there was no financial incentive. Based on the definition of adolescence proposed by the United Nations, youths aged 10–19,[23] only survey respondents aged 19 and younger were included for this paper. Due to an ethical-legal requirement of written parental consent for participants younger than 15 (The Act concerning the Ethical Review of Research Involving Humans[24]), which would complicate survey distribution, we opted to exclude those younger than 15 years. Therefore, the first survey item after consent concerned age and selecting *14 or younger* triggered information about exclusion and survey termination. The Checklist for Reporting Results of Internet E-Surveys[25] was used to report this study.

### Data collection

An anonymous questionnaire covering 83 items was designed in English and translated into Swedish. The questionnaire included a mixture of close-ended and open-ended questions with various response options (yes/no, multiple choice and free-text form). The design and selection of items were informed by prior studies.[1 18 26] Answering close-ended questions was mandatory, while free-text questions were optional. The Swedish survey was distributed over 12 pages and some items were conditionally displayed based on responses to other items. Including conditional questions, the number of items ranged from 1 to 10 items per page. Respondents could

1. How often do you think that you have accessed your health record during the last 12 months?

2. Did any of the following encourage or remind you to read your health record? (check all that apply)

3. Please indicate how much you disagree or agree with the following statement: I read my health record online… [reasons for reading]

4. How useful would it be to have access to the following information in the portal?

5. How useful would it be to have access to the following functions in the portal?

**Figure 3**  Survey items included in the study.

move back and forth between survey sections, however, they did not have an option to review the filled form before submission. For this study, five items were included (see figure 3) based on the study's aim. Three of these items used a 5-point Likert scale (ranging from disagree to agree), and two were close-ended questions. Data on use frequency were collected to enable comparison across groups. The items used in the current study are found in figure 3, and answer options and item number can be found in online supplemental appendix A (English original and Swedish translation). The full survey and further details about the data collection procedure are found in a separate publication.[27] It should be noted that the availability of information and functions in *Journalen* varies across regions in Sweden, which may affect respondents' assessment. However, the items on perceived usefulness are formulated anticipatory (see figure 3).

### Statistical analysis
Only completed questionnaires were analysed. Aside from using descriptive statistics for calculating percentages for different response options, the Fisher's exact test was used to explore the difference in use frequency based on whether an HCP (physician, nurse, psychologist, physiotherapist or other medical staff) had encouraged PAEHR use. Non-parametric tests were conducted due to the small sample size, and the significance level was set to 95%. The use frequency response options *This is my first time* and *2–9 times* as well as *10–20 times* and *More than 20 times* were merged for analysis. The SPSS V.28 software was used for analyses, and figures were produced using R.

### Patient and public involvement
The survey was tested with the public before launch to assess its clarity and response burden, leading to minor adjustments. There was no other patient and public involvement in the design or conduct of the study.

### RESULTS
### Characteristics of the participants
During the study period, 23 855 users opened the survey, and 15 867 (66.5%) started filling it in. Of these, 13 010 (82.0%) submitted the survey. Two (2/13 010, 0.02%) were 14 years old or younger and were excluded from the survey. Of the 13 008 respondents who completed

the survey, 218 were 15–19 years old (1.7%). Of respondents aged between 15 and 19, 77.1% (168/218) identified as women, 17.9% (39/218) as men while 5.0% (11/218) selected Other. See data on survey completion in a previous publication[27] and descriptive statistics for all variables in online supplemental appendix B.

### Experience with accessing records online
Two out of five respondents had logged in more than 20 times (85/218, 39.0%) and one-third had logged in 2–9 times (74/218, 33.9%) (see table 1).

### Encouragement to read the health records online
One survey question focused on whether respondents had been encouraged or reminded by anyone or any media source to read their records online. The most common response was *nobody encouraged me* (118/218, 54.1%) (see figure 4). Of four respondents that chose *Other*, one respondent provided a free-text description, citing 'myself'. Among the 47 individuals (47/218, 21.6%) who had been encouraged by some type of HCP, 36 (76.6%) were frequent users of the patient portal, which is higher than those who had not been encouraged (97/171, 56.7%) (p=0.018).

### Reasons for reading the health records online
Four reasons were chosen by more than half of respondents: *out of general curiosity* (170/218, 78.0%), *to get an overview of one's medical history and/or treatment* (159/218, 72.9%), *to ensure having understood what the physician or nurse said* (149/218, 68.3%) and *to improve one's understanding for one's health issue* (138/218, 63.3%) (see figure 5). Of 50 respondents who selected *Other*, 16 provided free-text comments. The comments included ensuring accuracy, memory reinforcement, poor notification frequency (ie,

**Table 1**  Participants' frequency of accessing their health records online

| Frequency of health record access, n (%) | Participants (n=218) |
|---|---|
| This is my first time | 11 (5) |
| 2–9 times | 74 (33.9) |
| 10–20 times | 48 (22) |
| More than 20 times | 85 (39) |

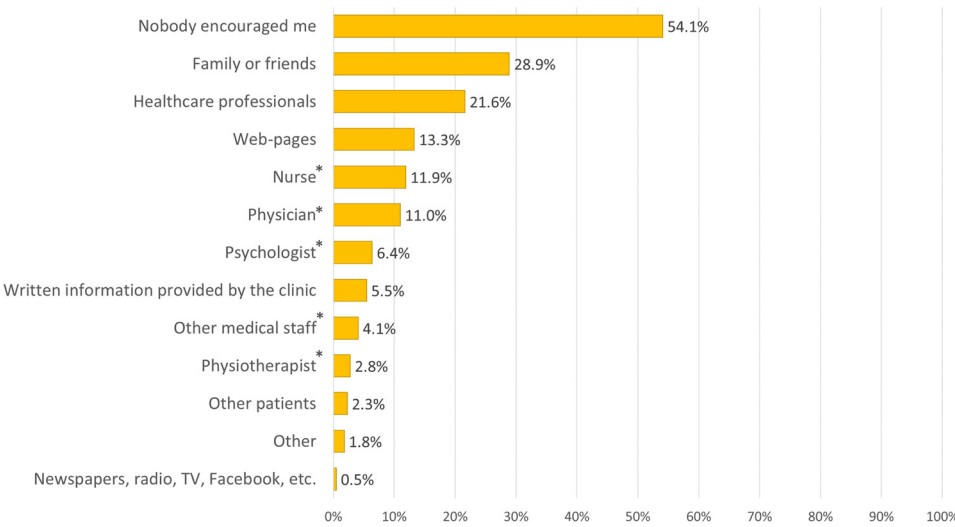

**Figure 4** Encouragement to read health records online. Because respondents could select multiple items, the total number can exceed 100%. *Types of HCPs.

logging in to check for new information due to lack of notifications), specification of sought information (eg, test results, COVID-19 vaccinations) and retrieving a record copy for monetary activity compensation.

### Usefulness of portal information and function

All information types were considered useful by more than three-fourths of participants (see figure 6).

Perceived usefulness of various functions was also examined, most of which do not currently exist in *Journalen*. Most functions were perceived as useful by a majority of respondents (see figure 7). Yet, four functions were considered useful by less than half of respondents: *accessing information and managing services for children* and for *family members, contributing information about expectations for the healthcare visit* and *blocking specific clinical notes from certain HCPs/medical staff*. Of 24 participants who wrote a comment under *Other*, seven referred to a specific function. Comments concerned the ability to view radiology images (median=4.5), diagnosis (median=3.5) and

dentistry records (median=5); ability to read children's records when under 18 (median=5) and records from childhood (median=5); and to view appointments from gynaecological care (median=5). Other free-text answers were non-responses or unspecified. *Medical certificates and other certificates* concerned certificates from HCPs, such as COVID-19 vaccination and for insurance and sick leave.

### DISCUSSION

In our national survey of PAEHR experiences in Sweden, adolescents perceived all proposed clinical information as consistently useful. Ratings for portal functions varied. Common reasons for reading records were curiosity and to get an overview of medical history. Sharing clinical information was the least common reason for accessing the record. Encouragement by HCPs to read records was associated with higher use. The reasons for reading

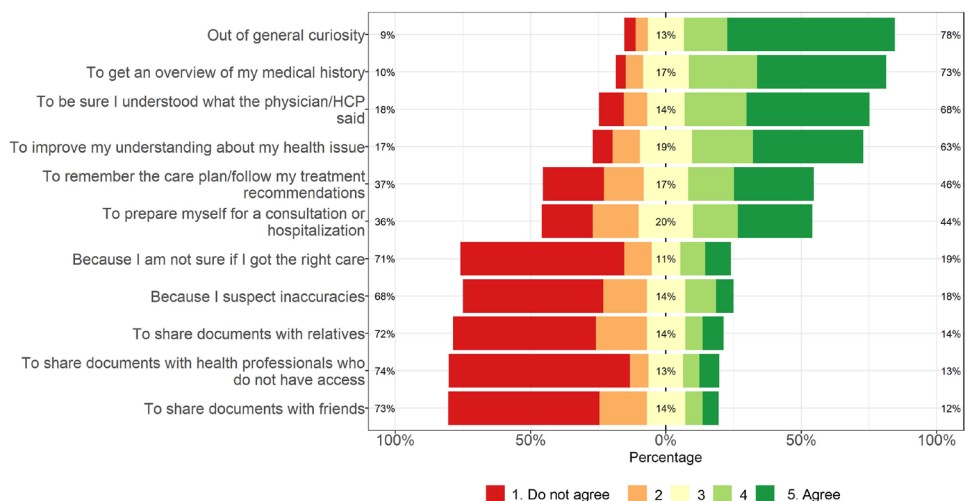

**Figure 5** Adolescents' reasons for reading health records online. HCP, healthcare professional.

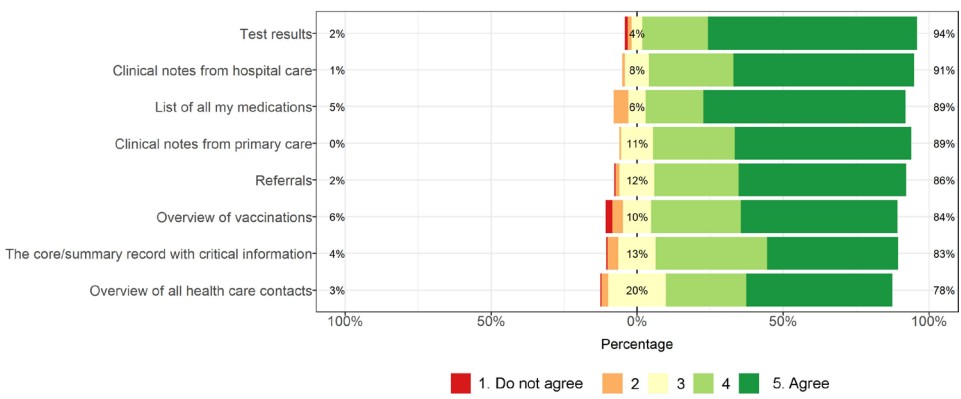

**Figure 6** Adolescents' perceived usefulness of different types of patient portal information.

records align with previous research on adults[18] and US adolescents.[21]

While all information types were rated highly on usefulness, test results were considered the most useful. This result accords with earlier research on adolescents,[15 19 20] but also with research on parents[28 29] and general adult populations.[18] Unsurprisingly, adolescents rated the ability to manage family members or children low. The function perceived as the third most useful was the ability to order and manage various certificates. This was likely associated with the timing of survey distribution, as the ongoing COVID-19 pandemic required Swedish citizens to present COVID-19 certificates for travelling and attending events. The boost of PAEHR use as a result of the COVID-19 pandemic has been mentioned previously.[30] Related to this, it is somewhat unexpected that immunisations were rated among the least useful types of information. However, Swedish regions differed in managing COVID-19 vaccination through the national e-service or other apps, and not all regions displayed information on vaccinations. Another

finding was that adolescents do not read their PAEHRs to share information. This finding is consistent with a previous survey study based in Canada, where adolescents without access to their records anticipated not being likely to share information from the records with others.[31]

Encouragement from HCPs was linked to higher use, indicating the influence of social factors in eHealth adoption.[32] Another possible explanation is that those with more active medical issues needed to review their health record more frequently. Encouragement from HCPs and family/friends has increased PAEHR use in adults,[33] suggesting that this may be an important factor influencing adoption and use also in adolescents. Lack of HCP encouragement aligns with research on adults in the USA[34] and Sweden,[35] and may stem from concerns about medical literacy, increased workload and uncertainty about visible information.[2 5 36–38] Not encouraging adolescents may lead to a missed opportunity for early patient involvement.[16] Increased PAEHR usage among adolescents can motivate policy-making stakeholders

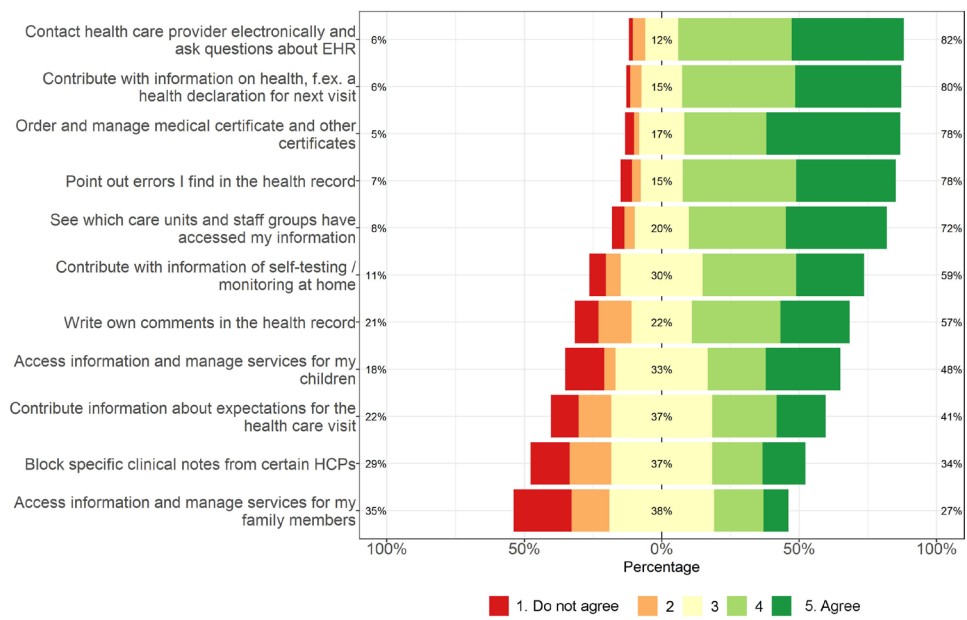

**Figure 7** Adolescents' perceived usefulness of patient portal functions. EHR, electronic health record; HCP, healthcare professional.

and EHR vendors to improve patient portals for this population.[5]

This study has limitations. Response biases in self-report surveys are unknown. Survey availability on the national patient portal may have influenced respondents' decisions to respond based on their competence and skills in using the service, potentially limiting the generalisability of the findings and causing positive bias as most adolescent respondents were frequent users. However, we opted to use convenience sampling as part of an exploratory approach. The majority of participants were women, which is consistent with adult and previous adolescent PAEHR users but represents a strong gender imbalance. Only 1.7% of the total respondents were aged 15–19, possibly due to younger individuals being less affected by health conditions and less engaged in personal health management. The survey was also designed for adults, and adolescents may have struggled to understand some questions. Furthermore, the reliability and validity of the survey used in this study have not been formally evaluated. While the survey was designed to capture the intended data, a comprehensive assessment of its psychometric properties was beyond the scope of this research. Still, the survey was tested with the public, as stated under the Patient and public involvement section.

The present results provide a broader understanding of adolescents reading their EHRs, however, further work is required to establish the viability of the findings in other settings. Several questions remain, relating to, for example, eHealth literacy and confidentiality. It should be noted that the current study included only a subset of items from a larger survey, and there is an intention to publish additional findings on the adolescent population, on participants' views about security and privacy. Regarding our research methodology, we offer a few suggestions for future work. First, frequency of use was defined by how many times (first time, 2–9 times, etc) the users accessed their records in the past 12 months. Another approach is to ask how often users access their records (daily, weekly, etc), which allows for an understanding of the visit regularity. Second, further research may benefit from providing a clearer distinction between a specific PAEHR service and patients' ORA more broadly. Lastly, including a larger and more evenly gender-distributed sample and specific patient groups with chronic diseases could furnish greater insights into adolescent users' experiences.

**Author affiliations**

[1] Department of Women's and Children's Health, Uppsala University, Uppsala, Sweden
[2] MedTech Science & Innovation Centre, Uppsala University Hospital, Uppsala, Sweden
[3] Beth Israel Deaconess Medical Center, Harvard Medical School, Boston, Massachusetts, USA
[4] School of Business, Örebro University, Örebro, Sweden
[5] Department of Information Technology, Uppsala University, Uppsala, Sweden
[6] School of Informatics, University of Skövde, Skövde, Sweden

**Contributors** JH wrote the manuscript, conducted the analyses, created the tables, designed the figures, and serves as the guarantor for the overall content. MH, CB, IS, JM, ÅC and HR contributed to the study design. All authors contributed to analysis and interpretation of the data, read the manuscript and provided feedback, and approved the final draft for submission.

**Funding** This work was supported by NordForsk (grant number: 100477) and the Swedish Research Council for Health, Working Life and Welfare (FORTE) (grant number: 2020-01229).

**Patient and public involvement** Patients and/or the public were involved in the design, or conduct, or reporting, or dissemination plans of this research. Refer to the Methods section for further details.

**Patient consent for publication** Not applicable.

**Ethics approval** This study involves human participants and was approved by the Regional Ethical Review Board in Uppsala, Sweden (EPN 2021/05229). Participants gave informed consent to participate in the study before taking part.

**Provenance and peer review** Not commissioned; externally peer reviewed.

**Data availability statement** Data are available upon reasonable request.

**ORCID iD**
Josefin Hagström http://orcid.org/0000-0003-2835-0259

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
