## [Reviewer comments · BMJ Paediatrics Open]

ARTICLE DETAILS

TITLE (PROVISIONAL)	Adolescents' reasons for accessing their health records online, perceived usefulness, and experienced provider encouragement: A national survey in Sweden
AUTHORS	Hagström, Josefin Blease, Charlotte Scandurra, Isabella Moll, Jonas Cajander, Åsa Rexhepi, Hanife Hägglund, Maria

VERSION 1 – REVIEW

REVIEWER	Dr. Teena Mary Joy Amrita Institute of Medical Sciences and Research Centre, Community Medicine
REVIEW RETURNED	29-Sep-2023

GENERAL COMMENTS	1. How did the participants of the survey come to know about the survey?2. Were the participants given an option to check or review the filled form before submission? If yes, how was it done?3. How were duplicate entries managed?4. Any comments regarding the non-representative sample?5. For the item in survey, 'encouragement to read health records online', were the participants allowed to choose multiple options?5 A. If yes then, is doing a Fischer's exact test in that result correct?5 B. If no then, the denominator exceeds 100%. Kindly check the item.6. For an international audience, explaining 177.se (Vårdguiden) a bit more would be beneficial.(Line number 23)7. If the 13 and 14 year age groups are being excluded, then why were they redirected for survey completion? (line number 36)
--

REVIEWER	Dr. Marianne Sharko Weill Cornell Medicine
REVIEW RETURNED	04-Oct-2023

GENERAL COMMENTS	This study explores the important topic of encouraging adolescents to access medical information online. It has some
--

	interesting findings on how frequently HCPs encourage or remind adolescents to read their health record and its association with use frequency. However, it is difficult to generalize the findings of the survey to represent all adolescents. The authors do not include the distribution of the ages of the respondents. In the United States we generally consider people 18 years and older to be adults. It is not clear how many of the respondents were 15-17 years old and considered adolescents in the US. In addition, this study appears to be written for an adult population without consideration of the literacy levels of adolescent respondents. For example, it is not clear that most 15- to 17-year-olds would understand what “provide a health declaration for next visit” refers to. While the authors generalize the findings of the survey to all adolescents, there is some selection bias in the participants of this survey. Since the participants took this survey through an online health portal, these are adolescents/young adults who already access an online portal and who choose to take surveys through the portal. There may be characteristics of the participants that make them more likely to do so, such as higher education, increased health needs, or older age. It is also possible that those who had more active medical issues may have been encouraged more frequently by their HCP to review their health record and that those with more active medical issues may have needed to review their health record more frequently, independent of HCP encouragement. The authors briefly refer to the challenges of protecting confidentiality in online medical information in the introduction; however, this study did not include any questions that ask about adolescent concerns of parental access to their medical information. The authors include in the Key Messages that “adolescents perceive all types of information in the portal useful”, however, it did not explore whether they would want all types of information to be accessible. We do not know if the patients who did not choose to open their portal, possibly for confidentiality concerns, would feel the same way about different types of health information. In the Key Messages, it is not clear what “educating HCPs on access to EHRS” is referring to. We do not know if the likelihood to encourage adolescents/young adults to read their health record is related to a lack of education.
--	--

	Possible typo on page 23, line 27 – refers to 4 functions considered less useful, but then lists only 3. Thank you for the opportunity to review this interesting study.
--	--

REVIEWER	Dr. Huang Chuan-Chin Brigham and Women's Hospital, Medicine
REVIEW RETURNED	09-Oct-2023

GENERAL COMMENTS	**Major concerns: The analytical plan in the method section and the results shown in Appendix C are confusing. It is not clear how a Fisher exact test can be used in a framework with a “check all that applies” setting. Please clarify and revise accordingly. **Minor comments: P17, Lines 17-18: We usually ensure consistency in the number of decimal places. P18, Line 23: Not sure what “8” means. P18, Line 33: Not sure what “9” means. P20, Line 24: It reads like that the reliability and validity of the survey have been evaluated. If this is the case, please add a sentence to emphasize it. P21, Lines 20-22: Please refer to my major concern. P21, Lines 34-60; Page 22, Lines 26-29: Decimal places changed frequently. Please make them consistent. Also, some started with numbers, followed by percentages, while others switched the order. This makes it difficult for the reader to follow. P21, Lines 43-44: Why were the numbers/percentages of 2-9 times not presented here? **Figures: Most of the figures had low resolution. Some text overlapped with the figure borders. Figure 4 used a comma, but not a period, for decimal separation, which is a bit unusual.
---

VERSION 1 – AUTHOR RESPONSE

Reviewer: 1

Dr. Teena Mary Joy, Amrita Institute of Medical Sciences and Research Centre

Comments to the Author

1. How did the participants of the survey come to know about the survey?

RESPONSE: Participants were informed about the survey upon login to their records. This is explained in the Methods, however we added some text for clarification (page 4, lines 13-15): Participants were notified about the survey when they logged into their records through the Swedish national PAEHR Journalen, by receiving a request to voluntarily participate in the survey, along with detailed information about the study.

2. Were the participants given an option to check or review the filled form before submission? If yes, how was it done?

RESPONSE: Thank you for your comment. Participants did not have an option to check or review the filled form before submission. We have added information about this in the Methods (p. 4, lines 33-34):

Respondents could move back and forth between survey sections, however did not have an option to review the filled form before submission.

3. How were duplicate entries managed?

RESPONSE: Thank you for pointing this out. Individuals received a request for participation when logging into the PAEHR, however the request disappeared once the individual had completed the survey. Multiple entries from the same individual was prevented in this way. We have added text to clarify this in the the CHERRIES checklist and the Method section (p. 4, lines 16-17):

The participation request did not appear once the patient had completed the survey.

4. Any comments regarding the non-representative sample?

RESPONSE: We thank the reviewer for pointing this out. We have added information about this in the Limitations (p. 6, lines 47-49):

Response biases in self-report surveys are unknown. Survey availability on the national patient portal may have influenced respondents' decisions based on their competence and skills in using the service, potentially limiting the generalizability of the findings and causing positive bias as most adolescent respondents were frequent users. However, we opted to use convenience sampling as part of an exploratory approach.

5. For the item in survey, 'encouragement to read health records online', were the participants allowed to choose multiple options?

RESPONSE: Thank you for this comment. We have now made a modification to the analysis. We recognize that this may have been unclear, and have modified for clarification (p. 4, lines 49-50): the Fisher's exact test was used to explore the difference in use frequency based on whether a HCP (physician, nurse, psychologist, physiotherapist, or other medical staff) had encouraged PAEHR use.

5 A. If yes then, is doing a Fischer's exact test in that result correct?

You are correct in that the multiple choice question should not be analyzed using a Fisher's Exact Test. We have modified the variable of encouragement to read to whether a HCP has encouraged the individual.

5 B. If no then, the denominator exceeds 100%. Kindly check the item.

RESPONSE: Since the respondents could select yes for multiple items, the total number could exceed 100%. This has been added to the figure caption (p. 7, line 5-6, Figure 4):

Because respondents could select yes for multiple items, the total number can exceed 100%.

6. For an international audience, explaining 177.se (Vårdguiden) a bit more would be beneficial. (Line number 23)

RESPONSE:

We have added information about this in the Introduction (p. 3, lines 18-19):

The e-service for PAEHRs in Sweden is Journalen, available via the national web patient portal 1177.se, accessible to patients in all 21 Swedish regions.8

7. If the 13 and 14 year age groups are being excluded, then why were they redirected for survey completion? (line number 36)

RESPONSE:

Thank you for this comment. To clarify, those aged 13 and 14 years old were redirected to a text explaining that they were being excluded because of being too young to participate. Thus, they did not technically complete the survey. Therefore, we have modified the text in the Methods as follows (p. 4, lines 22):

selecting 14 or younger triggered information about exclusion and survey termination.

Reviewer: 2

Dr. Marianne Sharko, Weill Cornell Medicine

Comments to the Author

This study explores the important topic of encouraging adolescents to access medical information online. It has some interesting findings on how frequently HCPs encourage or remind adolescents to read their health record and its association with use frequency.

However, it is difficult to generalize the findings of the survey to represent all adolescents.

The authors do not include the distribution of the ages of the respondents. In the United States we generally consider people 18 years and older to be adults. It is not clear how many of the respondents were 15-17 years old and considered adolescents in the US.

RESPONSE: We thank the reviewer for this point. We recognize that various definitions are used globally, yet we decided to use a definition of adolescents used by UN.

<https://www.un.org/esa/socdev/documents/youth/fact-sheets/youth-definition.pdf>

Regarding the following line in the manuscript (p. 5, lines 28-29):

Only survey respondents between 15-19 years old were included for this paper.

In addition, this study appears to be written for an adult population without consideration of the literacy levels of adolescent respondents. For example, it is not clear that most 15- to 17-year-olds would understand what "provide a health declaration for next visit" refers to.

RESPONSE: Thank you for this comment. We have added information about this in the Limitations (p. 6, lines 52-53):

The survey was also designed for adults, and adolescents may have struggled to understand some questions.

While the authors generalize the findings of the survey to all adolescents, there is some selection bias in the participants of this survey. Since the participants took this survey through an online health portal, these are adolescents/young adults who already access an online portal and who choose to take surveys through the portal. There may be characteristics of the participants that make them more likely to do so, such as higher education, increased health needs, or older age.

RESPONSE: We have added information about this in the Limitations, but have tried to emphasize this while remaining within the word limit (p. 6, lines 47-49):

Survey availability on the national patient portal may have influenced respondents' decisions based on their competence and skills in using the service, potentially limiting the generalizability of the findings and causing positive bias as most adolescent respondents were frequent users. However, we opted to use convenience sampling as part of an exploratory approach.

It is also possible that those who had more active medical issues may have been encouraged more frequently by their HCP to review their health record and that those with more active medical issues

may have needed to review their health record more frequently, independent of HCP encouragement.
RESPONSE: We have added information about this in the Discussion (p. 6, lines 37-38):
Another possible explanation is that those with more active medical issues needed to review their health record more frequently.

The authors briefly refer to the challenges of protecting confidentiality in online medical information in the introduction; however, this study did not include any questions that ask about adolescent concerns of parental access to their medical information.

RESPONSE: Thank you for this important comment. We recognize that protection of confidentiality is a key concern within implementation of access to pediatric EHRs. In fact, the full survey included questions on security and privacy, and we have recently submitted a separate manuscript detailing the results in relation to the adolescent respondents. Limitations of manuscript length and scope necessitated this separation of findings. In the Discussion, we have added information about our approach (p. 7, line 8):

It should be noted that the current study included only a subset of items from a larger survey, and there is an intention to publish additional findings on the adolescent population, on participants' views about security and privacy.

The authors include in the Key Messages that “adolescents perceive all types of information in the portal useful”, however, it did not explore whether they would want all types of information to be accessible. We do not know if the patients who did not choose to open their portal, possibly for confidentiality concerns, would feel the same way about different types of health information.

RESPONSE: We agree with this comment. We have added information about this in the Key Messages (p. 2, line 40):

Adolescents reading their records perceive all types of information in the EHR as useful.

In the Key Messages, it is not clear what “educating HCPs on access to EHRs” is referring to. We do not know if the likelihood to encourage adolescents/young adults to read their health record is related to a lack of education.

RESPONSE: Thank you for this comment. We are currently conducting a separate interview study that has indicated that this may be the case to a degree. However, we agree that the interpretation may be farfetched and we have edited the text by removing the text referring to education for HCPs (p. 2, lines 43-44):

Enhanced encouragement and support for adolescents on access to EHRs will be required to support the transition into adulthood and involve adolescents in their own care.

Possible typo on page 23, line 27 – refers to 4 functions considered less useful, but then lists only 3.

RESPONSE: Thank you for pointing this out. In the sentence, two functions (“accessing information and managing services for my children” and “accessing information and managing services for family members”) have been merged into “accessing information and managing services for my children and family members”. However, we understand that this is not sufficiently clear, and we have modified to clarify (p. 6, lines 3-5):

Yet, four functions were considered useful by less than half of respondents: accessing information and managing services for children and for family members, contributing information about expectations for the health care visit, and blocking specific clinical notes from certain HCPs/medical staff.

Thank you for the opportunity to review this interesting study.

RESPONSE: Thank you for providing useful and interesting comments.

Reviewer: 3

Dr. Huang Chuan-Chin

Comments to the Author

****Major concerns:**

The analytical plan in the method section and the results shown in Appendix C are confusing. It is not clear how a Fisher exact test can be used in a framework with a “check all that applies” setting. Please clarify and revise accordingly.

RESPONSE: Thank you for this comment. We have now made a modification to the analysis. We recognize that this may have been unclear, and have modified for clarification (p. 4, lines 48-50): the Fisher’s exact test was used to explore the difference in use frequency based on whether a HCP (physician, nurse, psychologist, physiotherapist, or other medical staff) had encouraged PAEHR use.

In the study aim (p. 4, lines 5-6):
and the association between frequency of use and HCP encouragement to read the EHR.

In the abstract (p. 2, line 12):
while Fisher’s exact test was used to explore differences in use frequency based on having been encouraged to read by a HCP.

In the discussion (p. 6, lines 19.20):
Encouragement by HCPs to read records was associated with higher use.

****Minor comments:**

P17, Lines 17-18: We usually ensure consistency in the number of decimal places.

RESPONSE: We appreciate this comment. The reason for the variety of decimal places is that we based the number on the sample sizes that the number is referring to (2 decimal places for numbers ≥ 1000 , 1 decimal place for numbers ≥ 100 , and so on). After looking at some other papers from the publication we have made the numbers consistent to 1 decimal place.

P18, Line 23: Not sure what “8” means.

RESPONSE: We thank you for pointing out this error - the “8” is a reference and should be in superscript. We have now edited this.

P18, Line 33: Not sure what “9” means.

RESPONSE: Again, thank you for pointing out this error - the “9” is a reference and should be in superscript. We have now edited this.

P20, Line 24: It reads like that the reliability and validity of the survey have been evaluated. If this is the case, please add a sentence to emphasize it.

RESPONSE: Thank you for your inquiry. We appreciate your interest in the reliability and validity of our survey instrument. However, it’s important to clarify that the reliability and validity of the survey have not been evaluated in this study. We acknowledge the need to provide this information to ensure the credibility of our research.

To address this concern, we will include the following statement in the paper to provide clarity on the evaluation of the survey’s reliability and validity under Limitations (p. 6, line 54, and p. 7, lines 1-3): Furthermore, the reliability and validity of the survey utilized in this study have not been formally evaluated. While the survey was designed to capture the intended data, a comprehensive assessment of its psychometric properties was beyond the scope of this research. Still, the survey was tested with the public before launch, as stated under Patient and public involvement.

P21, Lines 20-22: Please refer to my major concern.

RESPONSE: See response to earlier comment, this has been modified.

P21, Lines 34-60; Page 22, Lines 26-29: Decimal places changed frequently. Please make them consistent. Also, some started with numbers, followed by percentages, while others switched the

order. This makes it difficult for the reader to follow.

RESPONSE: Thank you for your feedback, and we appreciate your attention to detail. We understand the importance of consistency in reporting decimal places, and we have taken your comments into consideration to improve the clarity of our work.

We have addressed the issue of inconsistent decimal places, particularly in the case where the decimal was 0. To ensure clarity for our readers, we have made the necessary adjustment by including the decimal even when it is 0. This change has been implemented in the following passage (p. 5, line 32):

"Four reasons were chosen by more than half of respondents: out of general curiosity (170/218, 78.0%), to get an overview of one's medical history and/or treatment (159/218, 72.9%), to ensure having understood what the physician or nurse said (149/218, 68.3%), and to improve one's understanding of one's health issue (138/218, 63.3%).

P21, Lines 43-44: Why were the numbers/percentages of 2-9 times not presented here?

RESPONSE: Great comment, we have added this information (p. 5, lines 15-16): one third had logged in 2-9 times (74/218, 33.9%)

****Figures:**

Most of the figures had low resolution.

RESPONSE: Thank you for pointing this out. We have now improved the resolution of the figures.

Some text overlapped with the figure borders.

RESPONSE: Thank you for highlighting this. We have now modified the figures to remove overlapping of text with figure borders.

Figure 4 used a comma, but not a period, for decimal separation, which is a bit unusual.

RESPONSE: Thank you. This has been fixed.

VERSION 2 – REVIEW

REVIEWER	Dr. Teena Mary Joy Amrita Institute of Medical Sciences and Research Centre, Community Medicine
REVIEW RETURNED	12-Dec-2023

GENERAL COMMENTS	Thank you for addressing the comments. I am happy to have had an opportunity to review this interesting article.
--

REVIEWER	Dr. Huang Chuan-Chin Brigham and Women's Hospital, Medicine
REVIEW RETURNED	22-Dec-2023

GENERAL COMMENTS	The revised statement about the Fisher exact test remains unclear unless I missed something. The authors stated, 'One fifth (47/218, 21.6%) had been encouraged by HCPs to read their records, and having been encouraged by HCPs was related to a higher usage frequency (p = 0.043).' Please provide clear raw data and a description of the group with which the 47/218 was compared.
--

REVIEWER	Dr. Marianne Sharko Weill Cornell Medicine
REVIEW RETURNED	24-Dec-2023

GENERAL COMMENTS	Thank you for your thoughtful responses to the comments on this submission. Additional concerns: It would be helpful to include the definition of adolescence used in this submission - UNICEF/WHO/UNFPA: 10-19 years of age. It would also be helpful to see the age distribution of the respondents – are they evenly distributed across the age span, or do they skew towards one direction? Thank you for acknowledging the importance of adolescent privacy concerns. It is impossible to discuss the “Views and Experiences of Adolescents Reading their Health Records Online” without including their privacy concerns. I would suggest either including the privacy findings from the full survey or narrowing the title of this submission to more accurately reflect the information included in this submission (reasons for usage, associations with encouragement, and preferred functions). The title should also include that this national survey took place in Sweden.
---

VERSION 2 – AUTHOR RESPONSE

Reviewer: 1

Dr. Teena Mary Joy, Amrita Institute of Medical Sciences and Research Centre

Comments to the Author

Thank you for addressing the comments. I am happy to have had an opportunity to review this interesting article.

RESPONSE: Thank you for your observant comments.

Reviewer: 2

Dr. Huang Chuan-Chin, Brigham and Women's Hospital

Comments to the Author

The revised statement about the Fisher exact test remains unclear unless I missed something. The authors stated, 'One fifth (47/218, 21.6%) had been encouraged by HCPs to read their records, and having been encouraged by HCPs was related to a higher usage frequency (p = 0.043).' Please provide clear raw data and a description of the group with which the 47/218 was compared.

RESPONSE: Thank you for this comment. Based on previous reviewer comments pointing out that the response options for the question on encouragement were multiple-choice, we changed the analysis into a comparison of those who had been encouraged by a HCP and those who had not. The 47/218 who had been encouraged by a HCP were compared with the 171/218 who reported not being encouraged by a HCP.

Furthermore, we have made a minor modification to have more equally sized groups and no cell count below 5, merging the groups This is my first time and 2 to 9 times instead of 10-20 times and More than 20 times. Furthermore, we have made a minor modification to have more equally sized groups and no cell count below 5, merging the groups This is my first time and 2 to 9 times instead of 10-20 times and More than 20 times.

See the figure in the response letter PDF file for the results of the Fisher's exact test.

We have added text to clarify this in the text (p. 5, lines 27-28):

About a fifth (47/218, 21.6%) had been encouraged by some type of HCP while the remaining (171/218, 78.4%) had not, and Fisher's exact test revealed that those encouraged by a HCP were more frequent users of the patient portal than those who had not ($p=.043$).

Additionally, three rows have been added to the table "Encouragement to Read" in Appendix B (see PDF).

Reviewer: 3

Dr. Marianne Sharko, Weill Cornell Medicine

Comments to the Author

Thank you for your thoughtful responses to the comments on this submission.

RESPONSE: Thank you for your valuable comments.

Additional concerns:

It would be helpful to include the definition of adolescence used in this submission - UNICEF/WHO/UNFPA: 10-19 years of age.

It would also be helpful to see the age distribution of the respondents – are they evenly distributed across the age span, or do they skew towards one direction?

RESPONSE:

We have added the definition to the manuscript (p. 4, lines 19-23):

Based on the definition of adolescence proposed by the United Nations (UN) (age 10-19) [24], only survey respondents aged 19 and younger were included for this paper. Due to an ethical-legal requirement of written parental consent for participants younger than 15 (The Act concerning the Ethical Review of Research Involving Humans[25]), which would complicate survey distribution, we opted to exclude those younger than 15 years.

In regards to your second question, our data does not, unfortunately, include detailed information on age distribution, since survey respondents reported their age by selecting an age span, and the group investigated in this study selected the option 15-19 years old.

Thank you for acknowledging the importance of adolescent privacy concerns.

It is impossible to discuss the "Views and Experiences of Adolescents Reading their Health Records Online" without including their privacy concerns. I would suggest either including the privacy findings from the full survey or narrowing the title of this submission to more accurately reflect the information included in this submission (reasons for usage, associations with encouragement, and preferred functions).

The title should also include that this national survey took place in Sweden.

RESPONSE: The findings related to adolescents' views on security and privacy have been submitted as a separate manuscript, so we cannot include them in the present study.

The title has been modified to: "Adolescents' reasons for accessing their health records online, perceived usefulness, and experienced provider encouragement: A national survey in Sweden".

VERSION 3 – REVIEW

REVIEWER	Dr. Huang Chuan-Chin Brigham and Women's Hospital, Medicine
REVIEW RETURNED	29-Jan-2024

GENERAL COMMENTS	Thank you for responding to my comment. However, the revised information still does not provide sufficient details on how the authors obtained a p-value of 0.043. I believe that a statement similar to the following is what the reader would expect to see: 'Among the 47 individuals who had been encouraged by some type of healthcare provider (HCP), X (Y1%) were frequent users of the patient portal, which is higher than those who had not been encouraged (Z/171, Y2%) (p=0.043).'
--

VERSION 3 – AUTHOR RESPONSE

RESPONSE: We appreciate the correction and thank you for your patience. We have modified the text to clarify (p. 5, lines 27-30):

Among the 47 individuals (47/218, 21.6%) who had been encouraged by some type of healthcare provider (HCP), 36 (76.6%) were frequent users of the patient portal, which is higher than those who had not been encouraged (97/171, 56.7%) (p=.043).

Also, we have modified the groups to the original merge (10-20 times and 20 times or more) since we believe it better reflects frequent use (p. 5, line 1):

The use frequency response options 10-20 times and More than 20 times were merged for analysis to represent more frequent use.

VERSION 4 – REVIEW

REVIEWER	Dr. Huang Chuan-Chin Brigham and Women's Hospital, Medicine
REVIEW RETURNED	12-Feb-2024

GENERAL COMMENTS	If the numbers the authors provided were correct, the p-value of a two-sided Fisher exact test for would be 0.0175 (0.018), not 0.043. Please verify whether the provided numbers were incorrect or if the p-value was calculated incorrectly. Although the inference remains unchanged, please double-check and revise accordingly.
--

VERSION 4 – AUTHOR RESPONSE

We thank you again for your patience. We now understand how the reviewer meant. Revisions have been made in the manuscript regarding this analysis in terms of groups, see Results (p. 5, line 29): Among the 47 individuals (47/218, 21.6%) who had been encouraged by some type of HCP, 36 (76.6%) were frequent users of the patient portal, which is higher than those who had not been encouraged (97/171, 56.7%) ($p=.018$).

And Methods (p. 5, line 1):

The use frequency response options This is my first time and 2-9 times as well as 10-20 times and More than 20 times were merged for analysis.

VERSION 5 – REVIEW

REVIEWER	Dr. Huang Chuan-Chin Brigham and Women's Hospital, Medicine
REVIEW RETURNED	25-Feb-2024
GENERAL COMMENTS	No further comments

VERSION 5 – AUTHOR RESPONSE

None